# Natural Color Fool: Towards Boosting Black-box Unrestricted Attacks

**Shengming Yuan**
shengming.yuan@outlook.com

**Qilong Zhang**
qilong.zhangdl@gmail.com

**Lianli Gao**\*
lianli.gao@uestc.edu.cn

**Yaya Cheng**
yaya.cheng@hotmail.com

**Jingkuan Song**
jingkuan.song@gmail.com

Center for Future Media, University of Electronic Science and Technology of China

## Abstract

Unrestricted color attacks, which manipulate semantically meaningful color of an image, have shown their stealthiness and success in fooling both human eyes and deep neural networks. However, current works usually sacrifice the flexibility of the uncontrolled setting to ensure the naturalness of adversarial examples. As a result, the black-box attack performance of these methods is limited. To boost transferability of adversarial examples without damaging image quality, we propose a novel Natural Color Fool (NCF) which is guided by realistic color distributions sampled from a publicly available dataset and optimized by our neighborhood search and initialization reset. By conducting extensive experiments and visualizations, we convincingly demonstrate the effectiveness of our proposed method. Notably, on average, results show that our NCF can outperform state-of-the-art approaches by **15.0%∼32.9%** for fooling normally trained models and **10.0%∼25.3%** for evading defense methods. Our code is available at https://github.com/VL-Group/Natural-Color-Fool.

## 1 Introduction

Deep neural networks (DNNs) have achieved excellent performance on a large number of tasks, such as image recognition [1, 2], medical diagnostics [3, 4], and autonomous driving [5]. However, with the rise of the field of adversarial attack [6, 7, 8, 9, 10, 11, 12], the robustness of state-of-the-art DNNs [2, 13, 14, 15, 16] is put in question. Adversarial examples thus reveal the vulnerability of DNNs and inevitably raise concerns about widely deployed models. To avoid potential risks, it is crucial to expose as many of the model's "blind spots" as possible at this stage.

Generally, scenarios of adversarial attacks can divide into white-box and black-box. For the white-box scenario [17, 18, 19, 20], all information about the target model is publicly available, such as the architecture and parameters. Although the attacker can easily fool them with human-imperceptible perturbations in this case, it is not practical in the real world since deployed models are usually opaque to unauthorized users. To overcome this limitation, numerous works begin to study the black-box attack [7, 21, 22, 23, 24, 25], which relies on the cross-model transferability of adversarial examples. Typically, most existing black-box attacks maintain image quality by constraining the $L_p$-norm of perturbation [26, 23, 27, 8]. However, the $L_p$-norm constraint in RGB color space is not an ideal indicator for measuring the perceptual distance between two images since adversarial

---

\*Corresponding author

36th Conference on Neural Information Processing Systems (NeurIPS 2022).

examples with a small perturbation size may still look unnatural to human eyes [19]. Thus the unrestricted attack [28, 29, 30, 31, 32, 33, 34] is proposed — replacing the traditional small $L_p$-norm perturbations with uncontrolled yet human-imperceptible ones, which is more practically meaningful. Among existing unrestricted attacks, the unrestricted color branch is a very promising direction. As demonstrated in previous works [30, 32, 29], only manipulating the color of an image is not noticeable to human eyes but can easily fool DNNs. This is mainly because global uniform changes in images are usually imperceptible, and DNNs are vulnerable to certain large-scale patterns.

However, current unrestricted color attacks usually sacrifice the flexibility of the uncontrolled setting to ensure the naturalness of adversarial examples: either rely on intuition [28, 29] and objective metric [32] or use relatively small changes [30, 31]. Consequently, the transferability of adversarial examples generated by these methods is limited.

To remedy this, we propose a **Natural Color Fool (NCF)** based on a more flexible perturbation space. Specifically, we build a color distribution library that contains diverse yet realistic color distributions sampled from the ADE20K dataset [35] for each semantic class. By borrowing the color mapping technique of [36], we utilize the library to generate a set of image variants with realistic color distribution for each clean image. Then we use the substitute model to select the most adversarial variant. To boost its transferability, we further introduce **neighborhood search** and **initialization reset** to optimize, thus obtaining the resulting adversarial example. Figure 1 illustrates the simplified pipeline for our method.

In summary, our main contributions are:

- We observe that current unrestricted color attacks lack flexibility, which results in limited transferability of the adversarial examples.

- To alleviate this issue, we propose a Natural Color Fool (NCF), which fully exploits color distributions of semantic classes in an image to craft human-imperceptible, flexible, and highly transferable adversarial examples.

- Extensive experiments and visualizations demonstrate the effectiveness of our proposed method. Significantly, with high NIMA scores [37], NCF averagely outperforms state-of-the-art methods by **15.0%~32.9%** for fooling normally trained models, and **10.0%~25.3%** for evading defense models.

## 2 Related Works

### 2.1 Unrestricted Attacks

Unrestricted attacks [30, 38, 39, 32], which usually significantly modify the image but can yield human-imperceptible adversarial examples, have been widely studied in recent years. For example, some works modify domain-specific structural attributes of the image. SemanticAdv [38] uses an attribute-conditioned image editing model to modify the attributes of the face (e.g., modifying the degree of mouth opening). advCam [39] hides adversarial perturbations by using the style transfer technique (e.g., changing a traffic sign to a rust style). Of course, there are also many works modify generic attributes of images, such as texture and color. Specifically, tAdv [32] infuses the texture of another image to generate adversarial examples. EdgeFool [33] craft adversarial perturbations by enhancing edge details of an image.

Compared with texture-based attacks, color-based ones usually yield more natural results since manipulating groups of pixels along dimensions is less perceptible to human eyes. To our knowledge, Hosseini&Poovendran [28] first proposed the Semantic Adversarial Examples (SAE) in this branch. Specifically, they convert the image from the RGB color space to the HSV one, then uniformly and randomly perturbed the H (Hue) and S (Saturation) channels of the entire image. Laidlaw&Feizi [30] also notice that RGB color space is not a perceptually uniform space. Thus, they operate in the CIELUV color space and propose a ReColorAdv that uses a specific function to uniformly transform all pixels belonging to the same color. Similar to ReColorAdv, ACE [31] proposes a simple piecewise-linear color adjustment filter to manipulate the image. The main difference is that ACE treats three channels of an image independent of each other when transformed. AGV [40] increases the attack effectiveness by filter composition. Unlike them, cAdv [32] performs color transformation using a

pretrained colorization network and jointly varies input hints and masks to manipulate adversarial examples.

Among existing unrestricted color attacks, ColorFool [29], which exploits image semantics to modify colors selectively, is the most related to ours. Nevertheless, there are several significant differences between our work and ColorFool. First, ColorFool requires manually selecting several human-sensitive semantic classes (i.e., sky, water, plants, and humans), while our approach does not need. Second, ColorFool adds uncontrolled perturbations on human-insensitive semantic classes, while our approach utilizes a color distribution library as a guide. Third, ColorFool optimizes adversarial examples only via multiple random trials, while our approach also utilizes the gradient of the substitute model to fine-tune results.

## 2.2 Adversarial Defenses

To mitigate the threat of adversarial examples, various defense methods have been proposed for the past few years. Generally, existing defense techniques can fall into two categories: adversarial training [41, 42, 43, 44] and input pre-process [45, 46, 47]. The former gains immunity to attacks by leveraging adversarial examples during the training phase. For example, Tramèr *et al.* [42] improve the black-box robustness by considering adversarial examples generated by other irrelevant models. Xie *et al.* [43] introduce feature denoising modules and train them on adversarial examples to build white-box robust models. Although adversarial training is usually the most robust strategy, it suffers from expensive training costs. To overcome this limitation, the input pre-process branch is designed to cure the infection of adversarial examples before feeding to DNNs. For example, Guo *et al.* [45] introduce several input transformation techniques (e.g., JPEG compression [45]) to recover from the adversarial perturbations. Xie *et al.* [48] propose R&P which mitigates the adversarial influence through random resizing and padding. Liao *et al.* [49] come up with a high-level representation guided denoiser (HGD) to reduce the effect of adversarial perturbations. Jia *et al.* [46] propose an end-to-end image compression model called ComDefend against adversarial examples. Naseer *et al.* [47] train a neural representation purifier model (NRP) that cleans adversarial perturbation based on the automatically derived supervision.

## 3 Methodology

### 3.1 Problem Formulation

Let $x$ be a clean image with true label $y$ and $\mathcal{F}_\theta(\cdot)$ denote the deep learning classifier with parameters $\theta$. Formally, unrestricted attacks aim to craft human-imperceptible adversarial perturbations (may be very large in RGB color space) for $x$ so that the resulting adversarial examples $x'$ can induce $\mathcal{F}_\theta(\cdot)$ to misclassify:

$$\mathcal{F}_\theta(x') \neq y, \qquad s.t. \ x' \text{ is natural.} \tag{1}$$

In this paper, we focus on a more challenging black-box scenario (compared to the white-box scenario), where the target model's information (e.g., parameters and structure) is inaccessible. Therefore, adversarial examples can only be crafted via the accessible substitute model $\mathcal{F}_\phi(\cdot)$ and rely upon their transferability to fool target models.

### 3.2 Color Distribution Library

Existing unrestricted color attacks tend to sacrifice their flexibility so that they can generate human-imperceptible adversarial examples. For example, ReColorAdv [30] requires constraining the perturbation to a relatively small range, which cannot take full advantage of the "unrestricted" setting; cAdv [32] enforces the color belonging to the low-entropy cluster to remain unchanged, which inevitably reduces the attack space; ColorFool [29] manually splits an image into two parts and adds controlled noises on the human-sensitive part, which largely depends on the authors' intuition (but it varies from person to person).

To remedy the above limitation, we borrow the idea of [50] to construct a "distribution of color distributions" (DoD) for different semantic classes based on the ADE20K [35] dataset. Specifically, for each class, DoD provides 20 different dominant distribution sets (represented by the color distri-

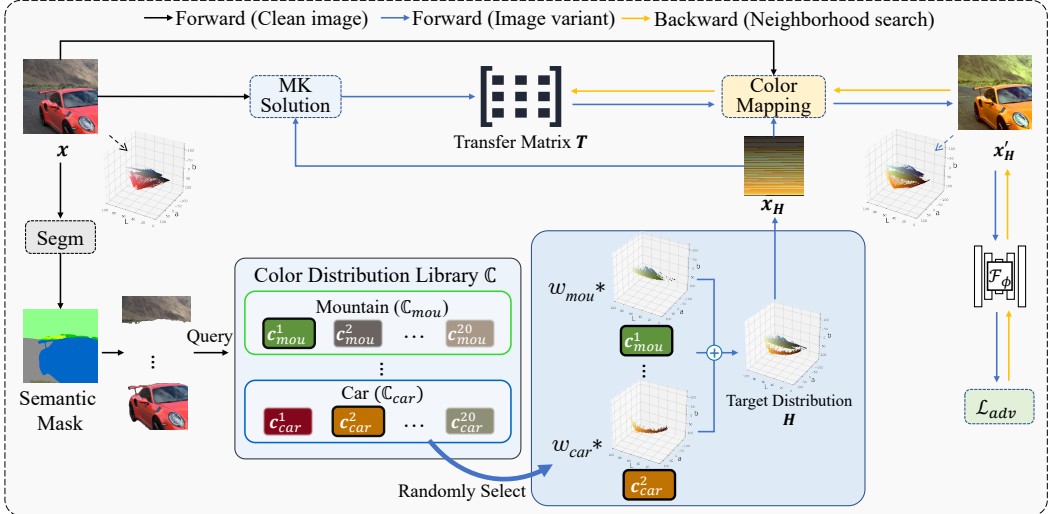

Figure 1: **The simplified pipeline of NCF** (optimizing one image variant without initialization reset). For the input image $x$, we first obtain its mask by a segmentation model (Segm). Then we randomly assign a realistic color distribution $c$ for each class via our color distribution library $\mathbb{C}$ and fuse them into a target $H$ based on the proportion of corresponding area $w$ in the image. Here $x_H$ is an image (without spatial information) directly converted by $H$ so that we can adopt Monge-Kantorovitch (MK) solution to obtain the transfer matrix $T$. With it and Eq. 4, we can get the color mapping image variant $x'_H$. Finally, we fine-tune $T$ (i.e., neighborhood search) according to the loss $\mathcal{L}_{adv}$ so that more threatening adversarial examples can be mapped.

butions), which are obtained by a hierarchical clustering based on color palettes[2] of corresponding segmentation class in the dataset.

Since the color distributions in each distribution set are similar and averaging all color distributions of a specific distribution set may not yield a natural representation, we randomly select a color distribution to represent the overall color characteristics for simplicity. Therefore, for a semantic class $\tilde{y}$, its realistic color distribution space can be expressed as:

$$\mathbb{C}_{\tilde{y}} = \{c_{\tilde{y}}^m\}_{m=1}^{20}, \tag{2}$$

where $c_{\tilde{y}}^m$ is a randomly sampled distribution from the $m$-th set, and our color distribution library $\mathbb{C}$ can be denoted as:

$$\mathbb{C} = \mathbb{C}_1 \cup \mathbb{C}_2 \cup ... \cup \mathbb{C}_M, \tag{3}$$

where $M$ denotes the total number of semantic classes contained in our library (see Appendix A for the pipeline of building color distribution library). In the following, we adopt this color distribution library to ensure the colors of adversarial examples are natural and coordinated, thus making our attack more flexible.

### 3.3 Natural Color Fool

To generate natural adversarial examples, precisely controlling the color perception is necessary. In this paper, we craft adversarial perturbations in CIELab color space where the perception is more uniform than RGB color space. The framework of our proposed **Natural Color Fool (NCF)** is illustrated in Figure 1 (refer to Appendix B for Algorithm of NCF). Formally, for a given image $x$, we first obtain a semantic mask through a semantic segmentation model. Then, for each semantic class $\tilde{y}$ in $x$, we randomly select a color distribution $c_{\tilde{y}}^i$ from $\mathbb{C}_{\tilde{y}}$ in the color distribution library. After that, we fuse the color distributions of all semantic classes based on the proportion of images occupied by each class (e.g., $w_{car}$, $w_{mou}$ in Figure 1), thus generating a target color distribution $H$. With it, we can use the following Eq. 4 to generate a target image variant $x'_H$ (refer to Figure 1)

---

[2]Color palette is a simplified color distribution composed of object's primary colors. It is used to reduce the computation cost of clustering.

which approximately replaces the color distribution of each semantic class of $x$ with corresponding one in target $H$ [36]:

$$x'_H = T(x - \mu_x) + \mu_{x_H}, \tag{4}$$

$$T\Sigma_x T^\top = \Sigma_{x_H}, \tag{5}$$

where $T \in \mathbb{R}^{3\times3}$ is the transfer matrix, $x_H$ is an image reconstructed by $H$ but without spatial information[3] (see Figure 1), and $\mu_x, \mu_{x_H}(\in \mathbb{R}^3), \Sigma_x, \Sigma_{x_H}(\in \mathbb{R}^{3\times3})$ are channel means of $x$, channel means of $x_H$, the covariance matrix of $x$, and the covariance matrix of $x_H$, respectively.

Formally, there are numerous solutions [51, 52, 53, 54] for $T$. However, most of them may not obtain an intended color mapping, i.e., only color proportions are expected. To tackle this issue, we follow Pitié *et al.* [36] and convert color mapping into a Monge–Kantorovich transportation problem, thus getting the solution for $T$:

$$T = \Sigma_x^{-1/2} \left( \Sigma_x^{1/2} \Sigma_{x_H} \Sigma_x^{1/2} \right)^{1/2} \Sigma_x^{-1/2}. \tag{6}$$

Intuitively, relying solely on randomly picking color distribution may limit the attack success rate. Since our color distribution library contains a wide variety of color combinations and we have accessible substitute models, there is no reason not to take advantage of them. Thus, we utilize the substitute model $\mathcal{F}_\phi(\cdot)$ to select a most adversarial distribution from $\eta$ different color distributions sampled from our color distribution library:

$$H^* = \arg\max_{H \in \mathbb{H}} \mathcal{L}_{adv}(\mathcal{F}_\phi(x'_H), y) \tag{7}$$

where $\mathbb{H}$ is the set containing $\eta$ different color distributions and $\mathcal{L}_{adv}$ is the C&W loss [18]:

$$\mathcal{L}_{adv}(z, y) = \max\{z_j : j \neq y\} - z_y, \tag{8}$$

where $z_j$ denotes the logit concerning the $j$-th class. To further boost our attack, we propose two techniques: **neighborhood search** and **initialization reset**. Specifically, the former aims to slightly adjust $T$ (derived from Eq. 6) to improve the attack success rate while ensuring a natural mapping result (see the comparison of adversarial examples with or without adopting neighborhood search in Appendix E). Thus, the neighborhood search can be formulated as the following optimization problem:

$$\arg\max_{T'} \mathcal{L}_{adv}(\mathcal{F}_\phi(T'(x - \mu_x) + \mu_{x_{H^*}}), y) \quad s.t. \ ||T - T'||_\infty \leq \epsilon, \tag{9}$$

where $\epsilon$ denotes the maximal perturbation size for $T$. In our paper, we adopt MI-FGSM [21] with $N$ iterations to optimize Eq. 9. Nonetheless, an adversarial transfer matrix $T'$ found in the neighborhood of a specific $T$ may not be effective, since the search space is indeed limited. To alleviate this issue, the initialization reset repeats the operation of selecting one from $\eta$ randomly sampled color distributions (i.e., Eq. 7) $K$ times, thus obtaining $K$ ideal initial color distributions for the neighborhood search. Then we choose the best one among them:

$$i' = \arg\max_{i \in 1,2,..,K} \mathcal{L}_{adv}(\mathcal{F}_\phi(T'_i(x - \mu_x) + \mu_{x_{H_i^*}}), y), \tag{10}$$

where $H_i^*$ and $T'_i$ denote the resulting color distribution and corresponding transfer matrix of the $i$-th reset. Consequently, resulting adversarial examples can be obtained by:

$$x' = T'_{i'}(x - \mu_x) + \mu_{x_{H_{i'}^*}}. \tag{11}$$

In Figure 2, we show resulting adversarial examples generated by our proposed method and recent works, i.e., cAdv [32], ColorFool [29], and ACE [31] (see Appendix G for more comparisons). Notably, despite our NCF implements significant modifications to the color distribution of the original images, the adversarial examples still look very natural.

---

[3]$x_H$ is used to calculate the covariance matrix. Please note that calculation result is the same no matter how we reconstruct it.

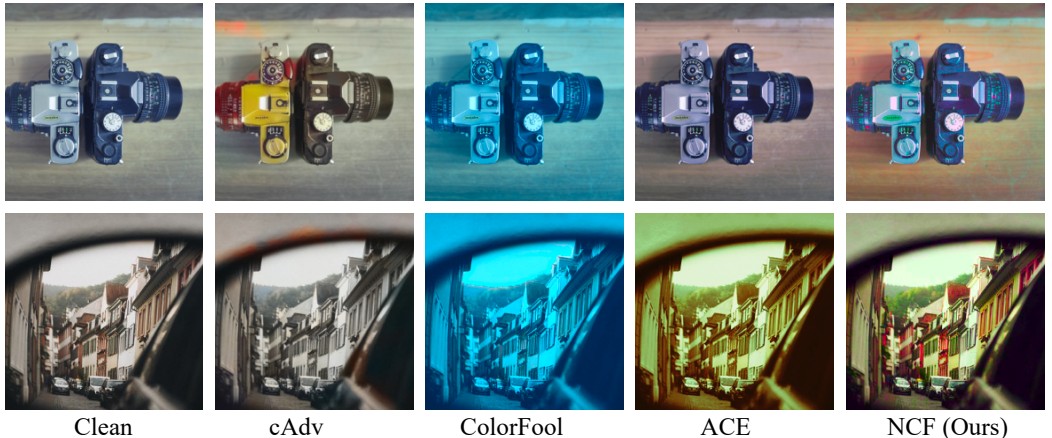

| Clean | cAdv | ColorFool | ACE | NCF (Ours) |

Figure 2: Adversarial examples generated by different unrestricted color attacks and ours.

## 4 Experiments

In this section, we first introduce the setup for our experiments in Section 4.1. Then we report attack success rates on normally trained and defense models in Section 4.2. After that, we compare the image quality of adversarial examples in Section 4.3. Finally, we conduct a series of ablation studies in Section 4.4. To better analyze the adversarial example, we also provide an illustration of attention shift [55] in Appendix F.

### 4.1 Setup

**Dataset.** We conduct our experiments on the ImageNet-compatible Dataset[4]. This dataset is comprised of 1,000 images and is widely used in recent transfer-based attacks [26, 22, 23, 56, 57, 58, 59].

**Models.** In this paper, we evaluate the vulnerability of various convolutional neural networks (CNNs) and vision transformers (ViTs). For CNNs, we consider both normally trained models and defense models. Normally trained models include VGG-19 [60], ResNet-18 (Res-18), ResNet-50 (Res-50) [2], Inception-v3 (Inc-v3) [61], DenseNet-121 (Dense-121) [13], and MobileNet-v2 (Mobile-v2) [62]. Defense methods contain JPEG [45], Gray (gray-scale conversion), HGD [49], R&P [48], ComDefend [46], NRP [47],[5] Inc-v3$_{ens3}$, IncRes-v2$_{ens}$ [42], Res152$_B$, Res152$_D$ and ResNeXt$_{DA}$ [43]. For ViT, we consider normally trained ViT-S/16 (ViT-S) [63], XCiT-N12/16 (XCiT-N12) [64], and DeiT-S [63].

**Implementation Details.** In all experiments, the semantic segmentation model is Swin-T [65] (the influence of different segmentation models on the results can be found in Appendix H) and the color distribution library size $M = 150$, the number of random searches $\eta = 50$, the iteration of neighborhood search $N = 15$, the maximum perturbation of transfer matrix $\epsilon = 0.2$, the step size $\alpha = \epsilon/N \approx 0.013$, the momentum $u = 0.6$ (for MI-FGSM), and the reset number $K = 10$ (see Appendix C for ablation of $K$). We compare our proposed method with Semantic Adversarial Examples (SAE) [28], ReColorAdv [30], cAdv [32], ColorFool [29], and ACE [31]. The parameters of each competitor follow the corresponding default setting. Our experiments are run on an NVIDIA TITAN Xp GPU with 12GB of memory.

---

[4]https://github.com/cleverhans-lab/cleverhans/tree/master/cleverhans_v3.1.0/examples/nips17_adversarial_competition/dataset.

[5]For HGD and R&P, we adopt the official models used in corresponding papers. For JPEG, Gray, ComDefend, and NRP, we adopt VGG-19 as the target model.

## 4.2 Transferability Comparison

### 4.2.1 Results on Normally Trained Models

In this section, we compare our proposed method with SA [28], ReColorAdv [30], cAdv [32], ColorFool [29] and ACE [31] on a variety of normally trained CNNs and Vision Transformers (ViTs). Adversarial examples are crafted via Res-18, VGG-19, Mobile-V2, and Inc-V3, respectively. For the results of ensemble attack [21] and using ViTs as the substitute model, please refer to Appendix I and D.

The attack success rates are presented in Table 1. We can observe that our resulting adversarial examples usually achieve the highest transferability compared to those generated via state-of-the-art competitors. For example, when the substitute model is Res-18, only 46.9%, 39.9%, and 40.8% of adversarial examples crafted by SA, ReColorAdv, and cAdv can successfully fool VGG-19. In comparison, our NCF can achieve a much higher transferability of **72.1%**. Besides, we also consider the transferability from CNNs to ViTs, as their structures are quite different. As shown in Table 1, when transferring adversarial examples from Mobile-V2 to XCiT-N12, recent ColorFool and ACE only obtain success rates of 22.8% and 22.6%, respectively, while our method can get **56.4%** success rate. On average, adversarial examples crafted by our method are capable of achieving a **54.5%** success rate, which significantly outperforms state-of-the-art approaches by **15.0%~32.9%**. This result convincingly demonstrates the effectiveness of our method in fooling normally trained models.

Table 1: **Transferability comparison on normally trained CNNs and ViTs**. We report attack success rates (%) of each method and the leftmost model column denotes the substitute model ("*" means white-box attack results).

| Model | Attacks | CNNs | | | | | | Transformers | | |
|---|---|---|---|---|---|---|---|---|---|---|
| | | Res-18 | VGG-19 | Mobile-v2 | Inc-v3 | Dense-121 | Res-50 | ViT-S | XCiT-N12 | DeiT-S |
| | Clean | 16.1 | 11.4 | 12.8 | 19.2 | 7.9 | 7.5 | 13.3 | 13.7 | 5.8 |
| Res-18 | SAE | 93.4* | 46.9 | 45.5 | 31.3 | 36.5 | 37.0 | 44.5 | 37.4 | 22.2 |
| | ReColorAdv | 98.6* | 39.9 | 47.3 | 38.2 | 37.2 | 38.1 | 21.4 | 36.7 | 17.3 |
| | cAdv | **100.0*** | 40.8 | 48.2 | 41.6 | 43.0 | 41.2 | 34.4 | 44.9 | 30.4 |
| | ColorFool | 93.0* | 27.8 | 30.5 | 28.1 | 19.8 | 22.9 | 35.5 | 22.3 | 9.2 |
| | ACE | 99.4* | 26.0 | 27.2 | 27.6 | 19.9 | 18.3 | 21.6 | 22.4 | 9.1 |
| | NCF (Ours) | 92.9* | **72.1** | **72.7** | **48.3** | **55.3** | **66.7** | **53.0** | **55.3** | **32.8** |
| VGG-19 | SAE | 52.2 | 91.4* | 48.8 | 32.3 | 39.3 | 39.0 | 48.3 | 37.6 | 24.3 |
| | ReColorAdv | 42.5 | 96.0* | 41.9 | 33.2 | 33.8 | 31.7 | 20.4 | 33.4 | 16.6 |
| | cAdv | 54.0 | **100.0*** | 48.0 | 43.7 | 43.4 | 40.7 | 38.8 | 43.9 | **32.9** |
| | ColorFool | 44.0 | 90.9* | 36.5 | 29.2 | 23.5 | 26.6 | 42.2 | 25.6 | 9.6 |
| | ACE | 33.4 | 99.7* | 27.8 | 28.3 | 21.6 | 18.0 | 20.7 | 21.6 | 9.5 |
| | NCF (Ours) | **73.7** | 93.3* | **70.3** | **49.4** | **53.6** | **64.3** | **56.5** | **53.5** | 30.7 |
| Mobile-V2 | SAE | 53.5 | 49.6 | 92.2* | 34.5 | 38.1 | 39.3 | 46.6 | 37.7 | 23.3 |
| | ReColorAdv | 46.3 | 36.5 | 97.8* | 36.4 | 32.4 | 34.4 | 20.7 | 36.7 | 20.0 |
| | cAdv | 54.7 | 39.9 | **100.0*** | 42.8 | 44.3 | 39.1 | 36.0 | 44.1 | 30.8 |
| | ColorFool | 41.5 | 30.6 | 93.2* | 28.1 | 23.3 | 24.5 | 39.7 | 22.8 | 9.4 |
| | ACE | 31.8 | 25.8 | 99.1* | 26.7 | 20.0 | 19.0 | 20.3 | 22.6 | 9.3 |
| | NCF (Ours) | **72.8** | **72.2** | 92.7* | **50.0** | **54.4** | **66.2** | **55.4** | **56.4** | **32.6** |
| Inc-V3 | SAE | 49.5 | 45.8 | 45.4 | 78.2* | 36.1 | 36.5 | **46.6** | 34.7 | 23.4 |
| | ReColorAdv | 26.5 | 19.9 | 21.9 | 96.2* | 17.2 | 15.9 | 16.3 | 22.5 | 10.5 |
| | cAdv | 32.7 | 23.4 | 27.6 | **99.8*** | 23.9 | 20.8 | 26.0 | 28.2 | 18.4 |
| | ColorFool | 40.4 | 31.8 | 35.4 | 84.1* | 23.9 | 25.3 | 42.6 | 26.5 | 12.6 |
| | ACE | 28.6 | 24.1 | 23.9 | 96.9* | 18.6 | 15.5 | 19.4 | 21.8 | 9.2 |
| | NCF (Ours) | **57.7** | **57.7** | **56.8** | 83.8* | **40.1** | **47.7** | 45.3 | **45.2** | **23.8** |

### 4.2.2 Results on Defense Models

Threats posed by $L_p$-norm adversarial attacks [7, 21, 23, 57] promote the emergence of adversarial defenses [41, 43, 47]. However, whether these defense mechanisms are robust against unrestricted attacks has not been extensively explored. Therefore, in this section, we consider both input pre-process defenses (i.e., JPEG [45], Gray, R&P [48], HGD [49], ComDefend [46], and NRP [47]) and adversarially trained models (i.e., Inc-v3$_{ens3}$, IncRes-v2$_{ens}$, Res$_B$ [42], Res$_D$, and ResNext$_{DA}$ [43]) to fully investigate the performance of our method.

Table 2: **Transferability comparison on defense CNNs**. We report attack success rates (%) of each method and the substitute model is Inc-v3.

| Methods | JPEG | Gray | R&P | HGD | ComDefend | NRP | $Res_B$ | $Res_D$ | $ResNeXt_{DA}$ | $Inc-v3_{ens3}$ | $IncRes-v2_{ens}$ |
|---------|------|------|-----|-----|-----------|-----|---------|---------|----------------|-----------------|-------------------|
| Clean | 17.4 | 26.7 | 2.4 | 2.0 | 28.2 | 16.8 | 29.2 | 24.4 | 19.8 | 7.7 | 4.2 |
| SAE | 52.7 | 33.6 | 13.0 | 11.5 | 62.2 | 53.7 | 55.6 | 53.5 | 50.1 | 19.8 | 13.7 |
| ReColorAdv | 26.9 | 34.4 | 7.8 | 5.6 | 38.0 | 27.6 | 32.9 | 28.8 | 24.3 | 14.7 | 9.3 |
| cAdv | 30.6 | 27.8 | 12.4 | 10.5 | 40.9 | 36.2 | 42.8 | 37.2 | 33.1 | 20.3 | 15.1 |
| ColorFool | 42.1 | 31.1 | 8.9 | 7.5 | 65.8 | 43.6 | 57.5 | 52.1 | 46.3 | 16.9 | 10.6 |
| ACE | 31.2 | 30.8 | 6.1 | 3.8 | 47.4 | 32.6 | 36.3 | 31.6 | 29.3 | 13.4 | 7.6 |
| NCF (Ours) | **62.6** | **37.9** | **26.0** | **23.5** | **73.5** | **66.2** | **61.2** | **58.2** | **60.1** | **32.6** | **26.9** |

The black-box success rates of unrestricted attacks are reported in Table 2. Similar to Table 1, our approach still consistently outperforms existing advanced methods by a large margin. On average, our proposed NCF is capable of making **48.1%** adversarial examples fool these defenses, while SA, ReColorAdv, cAdv, ColorFool, and ACE only obtain success rates of 38.1%, 22.8%, 27.9%, 34.8%, and 24.6%, respectively. Another interesting observation from Table 2 is that advanced defense algorithms are not necessarily effective when faced with unrestricted adversarial examples. For example, even white-box robust feature denoising models [43] are vulnerable, e.g., **60.1%** adversarial examples can fool ResNeXt$_{DA}$. We speculate the main reason is that the defense mechanism is intermediate feature denoising, while unrestricted color attacks mainly manipulate the color and do not generate significant high-frequency noise.

In addition, we notice that many input pre-process defenses with target model VGG-19 cannot defend against our proposed NCF, but instead boost the transferability of adversarial examples (compare to the VGG-19 results of Table 1). To better analyze this abnormal phenomenon, we visualize the purified images generated by JPEG, ComDefend, and NRP in Figure 3. Interestingly, we observe that all defenses have little effect on our unrestricted color perturbations. It is mainly because our adversarial examples are crafted using realistic colors as a guide and, therefore, can circumvent these defenses designed for common perturbation characteristics. Besides, we observe that JPEG and NRP suppress key features of "Great egret", and ComDefend generates many irrelevant repeating patterns. As discussed in [66], these side effects of defense methods can reduce the confidence of the model concerning the true class, resulting in our adversarial examples instead of boosting the transferability after being processed by these approaches. This result also reminds us that the robustness evaluation of defenses needs to be more comprehensive.

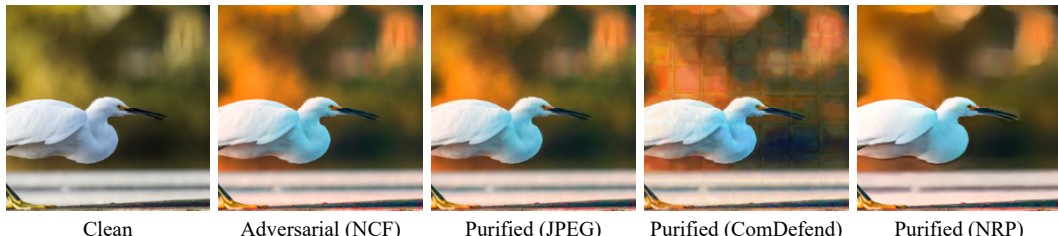

| Clean | Adversarial (NCF) | Purified (JPEG) | Purified (ComDefend) | Purified (NRP) |

Figure 3: **Visualization for the adversarial example and purified images**. This result shows that JPEG, ComDefend, and NRP cannot remove our unrestricted perturbation, but instead may distort key features (e.g., feather) of the object (the label is "Great egret"). Zoom in for better comparison.

## 4.3 Image Quality

Following ColorFool [29], we quantitatively evaluate image quality by a non-reference perceptual image quality measure called neural image assessment (NIMA) [37] which is highly correlated with human perception. In our paper, NIMA is based on the architecture of MobileNet [67] and is a composite of two models: one trains on AVA [68] for image technical assessment and another trains on TID2013 [69] for image aesthetics assessment. As reported in Table 3, image quality assessment results of our NCF are on par with those of state-of-the-art competitors. For NIMA (technical), both ACE and our NCF obtain a high score of over **4.939**, which is just **0.094** lower than clean images. The case of NIMA (aesthetic) is similar, where aesthetic scores of our adversarial examples and

clean images differ by only **0.097**, indicating that our NCF does not destroy the perceptual quality of images.

## 4.4 Ablation Studies

**About neighborhood search and initialization reset.** In this section, we investigate the effect of neighborhood search and initialization reset. Specifically, we use Inc-v3 to craft adversarial examples, and then evaluate the transferability towards Res-18, VGG-19, Mobile-v2, Dense-121, Res-50, ViT-S, XCiT-N12, and DeiT-S. The results are shown in Table 4. In terms of the white-box attack, NCF outperforms NCF-NS (NCF without NS), NCF-IR (NCF without IR), and NCF-IR-NS (NCF without IR and NS) by **17.7%**, **13.3%**, and **32.2%**, respectively. On the black-box attacks, neighborhood search and initialization reset bring an average performance improvement of **8.3%**, and **6.5%**. When both neighborhood search and initializa-

Table 3: **Image quality comparison.**

| Methods | NIMA ↑ (technical) | NIMA ↑ (aesthetics) |
|---|---|---|
| Clean | 5.033 | 4.457 |
| SAE | 4.924 | 4.326 |
| ReColorAdv | 4.886 | 4.117 |
| cAdv | 4.718 | 4.220 |
| ColorFool | 4.918 | **4.439** |
| ACE | **5.008** | 4.328 |
| NCF (Ours) | 4.939 | 4.360 |

tion reset techniques are used, the transferability towards black-box models can be improved by an average of **12.5%**. The significant performance gain confirms the effectiveness of neighborhood search and initialization reset. In addition, we notice that even if neither proposed technique is used, our method still surpasses state-of-the-art competitors, except for SAE. We attribute this phenomenon to the flexibility of our approach, which can naturally modify the whole image without restriction. Note that NCF-IR-NS does not mean selecting random colors to attack. The difference of them is discussed in Appendix J.

Table 4: **The effect of neighborhood search (NS) and initialization reset (IR).** Adversarial examples are crafted via Inc-v3 ("*" denotes white-box attack).

| Methods | Inc-v3* | Res-18 | VGG-19 | Mobile-v2 | Dense-121 | Res-50 | ViT-S | XCiT-N12 | DeiT-S |
|---|---|---|---|---|---|---|---|---|---|
| NCF | **83.8*** | **57.7** | **57.7** | **56.8** | **40.1** | **47.7** | **45.3** | **45.2** | **23.8** |
| NCF−NS | 66.1* | 49.3 | 47.2 | 46.9 | 31.2 | 39.1 | 41.7 | 34.4 | 18.1 |
| NCF−IR | 70.5* | 50.2 | 48.8 | 48.5 | 34.4 | 39.7 | 43.7 | 38.0 | 19.0 |
| NCF−IR−NS | 51.6* | 43.8 | 42.2 | 42.4 | 28.0 | 33.0 | 38.3 | 32.0 | 14.8 |

**About color distribution library size.** In this section, we investigate the influence of color distribution library size $M$ on attack success rates. Specifically, we tune $M$ from 0 to 150 with an interval of 15 and freeze other hyper-parameters, and adversarial examples are crafted via Inc-v3. Note that $M = 0$ means that our proposed library is not used, thus T is an identity matrix. The attack success rate is reported in Figure 4, where the dashed line indicates the white-box attack success rate and solid lines refer to the black-box ones. It can be observed that the library size $M$ plays a key role in attack success rates. Concretely, the attack success rates increase rapidly at first, then gradually stabilize when the library size exceeds 30, and reach the maximum at $M = 150$ in most cases. For instance, when $M = 150$, NCF has success rates of 57.7%, 57.6%, 56.8%, 40.1%, and 47.6% on Res-18, VGG-19, Mobile-v2, Dense-121, and Res-50, respectively. When $M = 0$, it significantly degrades to 24.6%, 18.6%, 21.1%, 14.8%, and 14.2%, respectively. Intuitively, this is because the target color distribution trivially turns to be the corresponding distribution of the original image, and thus the space we perturb on the original image will be reduced. This convincingly demonstrates the effectiveness of building a class-rich color distribution library.

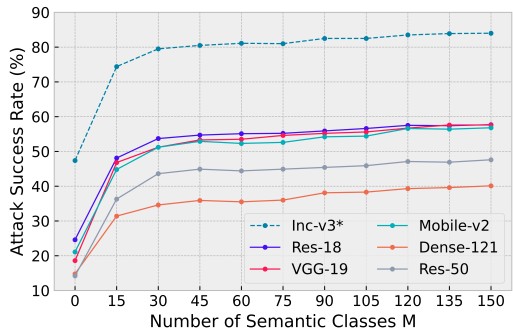

Figure 4: **Attack success rates (%) w.r.t. the number of semantic classes** $M$. The substitute model is Inc-v3.

# 5 Conclusion

In this paper, we propose a black-box unrestricted color attack called Natural Color Fool (NCF). Different from existing works which either rely on intuition and objective metrics or use relatively small changes to yield natural results, our method instead exploits color distribution to craft human-imperceptible, flexible, and highly transferable adversarial examples. Extensive experiments and visualizations demonstrate the effectiveness of our NCF. Notably, our adversarial examples can easily evade current $L_p$ robust defense techniques. Furthermore, the transferability of our adversarial examples is even boosted after being processed by several input pre-process defenses. We hope our work can draw researchers' attention to unrestricted color attacks.

**Limitation.** In our paper, for each semantic class, we provide 20 different color distributions for choosing. Therefore, this library is discrete, which may be insufficient to model the whole color space. In the future, we will try to build a continuous color space to expand the spatial range and more accurately simulate the color range of objects.

**Negative Societal Impacts.** Adversarial examples crafted by our proposed unrestricted color attack look very natural but are with strong black-box transferability even towards defenses. Therefore, unscrupulous people may adopt our method to undermine real-world applications, which inevitably raises new concerns about AI safety.

# 6 Acknowledgments

This study is supported by grants from National Key R&D Program of China (2022YFC2009903/2022YFC2009900), the National Natural Science Foundation of China (Grant No. 62122018, No. 62020106008, No. 61772116, No. 61872064).

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
