# OpenReview forum: "Natural Color Fool: Towards Boosting Black-box Unrestricted Attacks"
_NeurIPS.cc/2022/Conference — NeurIPS 2022 Accept_

### Official Review · Reviewer_Hmyg · 2022-07-10

**Rating:** 6
**Confidence:** 4
**Soundness:** 3 good
**Presentation:** 4 excellent
**Contribution:** 3 good

**Summary:**

This paper introduces a new unrestricted attack (NCF) that boosts the black box performance. It is done by color converting, neighborhood search, and initialization reset. The experiment includes both CNN and Vit architectures and compares a host of baseline methods. Overall, NCF achieves state-of-the-art performance.

**Questions:**

See previous section

**Strengths And Weaknesses:**


Strengths:
1. The paper specifically targeted on black-box setting and achieved much higher adversarial transferability than baseline methods.
2. The authors consider a lot of recent ViT models and lots of defense methods to evaluate.

Weaknesses:
1. The authors state that "current unrestricted color attacks lack flexibility, which results in limited transferability of the adversarial examples". First, I am not exactly sure what flexibility means here. Then, I observe (from table 1) that other attacks usually outperform NCF in the white-box setting. If NCF is more flexible, shouldn't it also have a higher white-box attack success rate? Authors should explain why NCF is better on transferability and worse on white-box attacks.

2.  I noticed that all black-box attacks are generated from one substitute model, whereas in some real-world settings, attackers may have multiple substitute models. Could the NCF and baseline methods improve if more substitute models are used?  This paper [1] is also worth to be discussed.

[1] Baia et al. One for Many: an Instagram-inspired black-box adversarial attack


========================================================================
After rebuttal:
I think the authors carefully address my concerns, and I am willing to increase my score.

---

> ### Author Response · Authors · 2022-08-02
> **Response to Reviewer Hmyg**
>
> _**Q1 What does the “flexibility” mean?**_
>
> **A1**: Please refer to the response to the generic comment.
>
> _**Q2  Why NCF is better on transferability and worse on white-box attacks.**_
>
> **A2**: NCF aims to generate coordinated, natural-looking adversarial examples. Instead of perturbing each pixel value individually, pixels of similar color are usually adjusted uniformly, i.e., color-wise not pixel-wise perturbation. Besides, to ensure the efficiency of NCF, our perturbation is optimized in only $N=15$ iterations. if we increase $N$ to 100, white-box success rate on res18 can be further improved by about 3\% (i.e., 92.9\% -> 95.1\%), which outperforms SAE and ColorFool (Please note ReColoradv, cAdv and ACE need more iterations, e.g.,the maximum iteration for ACE is 500). Therefore, the reported attack success rates on white-box models is limited. As for the black-box attack, we argue it is because our color-wise perturbation does not over-fit the white-box model and thus achieving higher black-box transferability (like FGSM vs. I-FGSM).
>
> Please note that our white-box results do not contradict the flexibility of our approach. Take ColorFool as an example. It needs to **manually** split an image into two parts and adds controlled noises on the human-sensitive part, which largely depends on the authors’ intuition (but it varies from person to person). By contrast, our NCF can **automatically** select an adversarial color distribution for each semantic class. In this case, "automatic" reflects the flexibility of NCF as opposed to the "manual" nature of ColorFool.
>
> _**Q3 Could the NCF and baseline methods improve if more substitute models are used? This paper [f] is also worth to be discussed.**_
>
> **A3**: Thanks for providing [f], which is an interesting paper and we have discussed it in our revision. As for ensemble model attack (fusing the logits of multiple models like [g]), here we report the result of NCF. As indicated in Table c, the attack success rate of NCF can be further improved when crafting via an ensemble of models.
>
> Table c: Comparison of ensemble attack and single model attack. We report attack success rates (\%) of NCF and the leftmost model column denotes the substitute model, where Ensemble means an ensmeble of Res-18, VGG-19 and Mobile-v2.
>
> | Models    | Dense-121 | Res-50 | ViT-S | XCiT-N12 | DeiT-S |
> |-----------|-----------|--------|-------|----------|--------|
> | Clean     | 7.9       | 7.5    | 13.3  | 13.7     | 5.8    |
> | Res-18    | 55.3      | 66.7   | 53.0  | 55.3     | 32.8   |
> | VGG-19    | 53.6      | 64.3   | 56.5  | 53.5     | 30.7   |
> | Mobile-V2 | 54.4      | 66.2   | 55.4  | 56.4     | 32.6   |
> | Ensemble  | **63.5**  | **71.6** | **59.7**  | **61.7**     | **37.0**   |
>
> [f] Alina Elena Baia, Alfredo Milani, and Valentina Poggioni. One for many: an Instagram inspired black-box adversarial attack. 2021.
>
> [g] Yinpeng Dong, Fangzhou Liao, Tianyu Pang, Hang Su, Jun Zhu, Xiaolin Hu, and Jianguo Li. Boosting adversarial attacks with momentum. In CVPR, 2018

---

> > ### Comment · Reviewer_Hmyg · 2022-08-05
> > **More comments**
> >
> > **Why NCF is better on transferability and worse on white-box attacks.**
> >
> > I am confused about what makes your attacks achieve high transferability, color-wise perturbation, or fewer attacking iterations. Therefore, I wonder what the black box transferability is after you use $N=100$.
> >
> > **Could the NCF and baseline methods improve if more substitute models are used?**
> >
> > Thanks for running the experiments.

---

> > > ### Author Response · Authors · 2022-08-06
> > > **Sincerely thanks for your response**
> > >
> > > Thank you a lot for the reply, which gives us the opportunity to address your concerns more clearly.
> > >
> > > _**Q4 What makes your attacks achieve high transferability, color-wise perturbation, or fewer attacking iterations.**_
> > >
> > > **A4:** Both color-wise perturbation and more attacking iterations can help to improve transferability. For the former, color-wise perturbation is similar to patch-wise perturbation [h] which has been demonstrated to improve transferability. For the latter, [i] has shown that existing transfer methods with more iterations yield better results. Table d also indicates that $N=100$ performs better than $N=15$.
> > >
> > >
> > > Table d: The effect of the iterations ($N$) of neighborhood search on the attack success rates (\%) of NCF. We fix the maximum perturbation of the transfer matrix and increase the number of iterations ("*" denotes white-box attack).
> > > | $N$ | Res-18*   | VGG-19   | Mobile-v2 | Inc_v3   | Dense-121 | Res-50   | ViT-S    | XCiT-N12 | DeiT-S   |
> > > | --- | --------- | -------- | --------- | -------- | --------- | -------- | -------- | -------- | -------- |
> > > | 15  | 92.9*     | 72.1     | 72.7      | 48.3     | 55.3      | 66.7     | 53.0     | 55.3     | 32.8     |
> > > | 100 | **95.1*** | **74.2** | **75.5**  | **50.9** | **57.7**  | **69.0** | **55.0** | **56.7** | **34.8** |
> > >
> > >
> > > _**Q5 Could the NCF and baseline methods improve if more substitute models are used?**_
> > >
> > > **A5:** Yes, as demonstrated in the following Table e, using more substitute models can improve the performance of NCF and baseline methods.
> > >
> > > Table e: Comparison of ensemble attack and single model attack.
> > >
> > > | Attacks    | Models       | Dense-121 | Res-50   | ViT-S    | XCiT-N12 | DeiT-S   |
> > > | ---------- | ------------ | --------- | -------- | -------- | -------- | -------- |
> > > |    SAE     | Res-18        | 36.5      | 37.0     | 44.5     | 37.4     | 22.2     |
> > > |            | VGG-19        | 39.3      | 39.0     | 48.3     | 37.6     | 24.3     |
> > > |            | Mobile-v2 | 38.1      | 39.3     | 46.6     | 37.7     | 23.3     |
> > > |            | Ensemble     | **44.2**  | **47.0** | **53.7** | **42.1** | **26.2** |
> > > | ReColorAdv | Res-18        | 37.2      | 38.1     | 21.4     | 36.7     | 17.3     |
> > > |            | VGG-19        | 33.8      | 31.7     | 20.4     | 33.4     | 16.6     |
> > > |            | Mobile-v2 | 32.4      | 34.4     | 20.7     | 36.7     | 20.0     |
> > > |            | Ensemble     | **47.2**  | **50.5** | **25.9** | **43.3** | **24.2** |
> > > | cAdv       | Res-18        | 43.0      | 41.2     | 34.4     | 44.9     | 30.4     |
> > > |            | VGG-19        | 43.4      | 40.7     | 38.8     | 43.9     | 32.9     |
> > > |            | Mobile-v2 | 44.3      | 39.1     | 36.0     | 44.1     | 30.8     |
> > > |            | Ensemble     | **59.9**  | **59.0** | **45.7** | **56.1** | **41.6** |
> > > | ColorFool  | Res-18        | 19.8      | 22.9     | 35.5     | 22.3     | 9.2      |
> > > |            | VGG-19        | 23.5      | 26.6     | 42.2     | 25.6     | 9.6      |
> > > |            | Mobile-v2 | 23.3      | 24.5     | 39.7     | 22.8     | 9.4      |
> > > |            | Ensemble     | **32.2**  | **36.8** | **49.5** | **30.0** | **14.1** |
> > > | ACE        | Res-18        | 19.9      | 18.3     | 21.6     | 22.4     | 9.1      |
> > > |            | VGG-19        | 21.6      | 18.0     | 20.7     | 21.6     | 9.5      |
> > > |            | Mobile-v2 | 20.0      | 19.0     | 20.3     | 22.6     | 9.3      |
> > > |            | Ensemble     | **29.2**  | **27.9** | **25.4** | **27.7** | **10.7** |
> > > | NCF (Ours)  | Res-18        | 55.3      | 66.7     | 53.0     | 55.3     | 32.8     |
> > > |            | VGG-19        | 53.6      | 64.3     | 56.5     | 53.5     | 30.7     |
> > > |            | Mobile-v2 | 54.4      | 66.2     | 55.4     | 56.4     | 32.6     |
> > > |            | Ensemble     | **63.5**  | **71.6** | **59.7** | **61.7** | **37.0** |
> > >
> > >
> > > [h] Lianli Gao, Qilong Zhang, Jingkuan Song, Xianglong Liu, and Heng Tao Shen. Patch-wise attack for fooling deep neural network. In ECCV 2020.
> > >
> > > [i] Zhengyu Zhao, Zhuoran Liu, Martha Larson. On Success and Simplicity: A Second Look at Transferable Targeted Attacks. In NeurIPS 2021.

---

> > > > ### Comment · Reviewer_Hmyg · 2022-08-06
> > > > **More comments**
> > > >
> > > > Thanks for addressing my concerns. I am willing to increase the score.

---

### Official Review · Reviewer_C6Jv · 2022-07-10

**Rating:** 7
**Confidence:** 4
**Soundness:** 3 good
**Presentation:** 2 fair
**Contribution:** 3 good

**Summary:**

The paper proposes a color based unrestricted adversarial black box attack on image classification deep neural networks by transferring the adversarial examples using a substitute network. Authors propose to generate a natural color distribution library based on the publicly available ADE20K dataset. They create a library of distinct color distributions for 150 semantic classes. They generate the adversarial examples by randomly picking several color distributions for each semantic class from the library and find the image that fools the substitute network. In addition, they perform neighborhood search on a 3x3 Transfer Matrix T that performs the color mapping to further boost the attack success rate. Furthermore, they reset the Transfer Matrix T like random restarts in PGD attack. Extensive results show that the proposed method maintains the image quality and boosts the attack transferability significantly compared to the existing methods. Major boost of the transferability comes from optimizing the matrix T to generate adversarial examples.

**Questions:**

In addition to the points mentioned in weakness, I have two additional questions:

At line 129, it is mentioned that “Since the color distributions in each distribution set are similar and averaging all color distributions of a specific distribution set may not yield a natural representation, we randomly select a color distribution to represent the overall color characteristics for simplicity”. If their distributions are similar, why not select a single color distribution as a template from each set? What does the natural representation mean here?

In line 141, authors mention that “we craft adversarial perturbations in CIELab color space where the perception is more uniform than RGB color space”. What does the perception is more uniform mean here? How does that help to create adv examples?


**Limitations:**

As per my understanding, authors briefly addressed the limitations and negative impact in their work.

**Strengths And Weaknesses:**

Strengths:
1)	Well written paper. Most of the parts are easy to understand.
2)	Proposes a novel method to generate transferrable adversarial attack.
3)	Method explanation is easy to follow.
4)	Conducted extensive experiments on wide variety of network architectures.
5)	Shown a significant improvement of the attack success rate with the proposed method on both undefended and defended models (L_p based defenses and input processing defenses).

Weakness:
1)	In the beginning of the paper, authors often mention that previous works lack the flexibility compared to their work. It is not clear what does it mean and thus makes it harder to understand their explanation.
2)	It is not clear regarding the choice of 20 distribution sets. Can we control the number of distribution sets for each class? What if you select only few number of distribution set?
3)	The role of Tranfer Matrix T is not discussed or elaborated.
4)	It is not clear how to form the target distribution H. How do you formulate H?
5)	There is no discussion on how to generate x_H from H and what does x_H constitute of?
6)	Despite the significant improvement, it is not clear how this proposed method boost the transferability of the adversarial examples.

---

> ### Author Response · Authors · 2022-08-02
> **Response to Reviewer C6Jv (1 of 2)**
>
> Thank you for your positive feedback and insightful comments. Please see our detailed response below.
>
> _**Q1 What does the "flexibility" mean?**_
>
> **A1**:Please refer to the response to the generic comment.
>
> _**Q2 It is not clear regarding the choice of 20 distribution sets. Can we control the number of distribution sets for each class? What if you select only few number of distribution set?**_
>
> **A2**: Yes, we can control the number of distribution sets for each class when building the color distribution library. But if we select only few number of distribution set for each class, the attack space for our method will be reduced and thus limiting the performance of our NCF. It is because that each set is represented by a single color distribution (i.e. one style) for simplicity (see lines 129-132). If the number of distribution set for each class is 1, the result will be the same no matter how many times initialization reset (IR) is executed. However, as shown in Table 4, IR plays a very important role in attack performance. Therefore, in this paper, we use 20 (instead of "few number") different distribution sets.
>
> _**Q3 It is not clear how to form the target distribution H. How do you formulate H?**_
>
> **A3**: $\pmb{H}$ denotes the overall color distribution consisting of the color distributions of all semantic classes in an image. Specifically, we randomly choose a color distribution (from the color distribution library) for each semantic class and weight the sum according to the area ratio $w$ of the relevant semantic classes to obtain the target color distribution $\pmb{H}$:
>
> $$\pmb{H}=\sum_{\tilde{y}=1}^{|\tilde{Y}|} w_{\tilde{y}} \cdot \pmb{c}_{\tilde{y}},$$
>
> where $\tilde{Y}$ denotes the semantic classes contained in the image, $w_{\tilde{y}}$ denotes the area ratio of semantic class $\tilde{y}$ in an image, and $c_{\tilde{y}}$ denotes the target color distribution chosen for semantic class $\tilde{y}$. Essentially, $\pmb{H}$ and $c_{\tilde{y}}$ are matrices of size $100\times256\times256$. If $c_{\tilde{y}}[L_i,A_i,B_i]=w\neq 0$, it means that in the current style, the semantic class $\tilde{y}$ contains $w*100$% of the pixels with the value $(L_i,A_i,B_i)$.
>
>
> _**Q4 There is no discussion on how to generate $\pmb{x_H}$ from $\pmb{H}$ and what does $\pmb{x_H}$ constitute of?**_
>
> **A4**: $\pmb{x_H}$ is an intermediate variable used for color transfer, which is reconstructed by $\pmb{H}$ but without spatial information. Specifically, $\pmb{x_H}$ is generated based on the image size and the color ratio recorded in $\pmb{H}$. It aims to make the color distribution of $\pmb{x_H}$ equal to the target color distribution $\pmb{H}$. The following is the pseudo-code for generating $\pmb{x_H}$ from $\pmb{H}$:
>
> ```python
> def convert(H, img_h, img_w):
>     """
>     Args:
>         H: target color distribution
>         img_h: height of the image to be attacked
>         image_w: width of the image to be attacked
>
>     """
>     img_area = img_h*img_w  # Image size
>     x_H = np.zeros(img_area, 3)  # Initialization
>     pos = np.nonzero(H)  # All index positions in H that are not zero
>
>     start = 0
>     for i in len(pos):
>         (L, A, B) = pos[i]  # Extracts the color
>         num = img_area*H[L, A, B]  # The number of pixels of the color (L,A,B) in x_H
>
>         x_H[start: start+num] = (L, A, B)
>         start = start + num
>
>      x_H = x_H.reshape(img_h, img_w, 3)
>
>  return x_H
>  ```

---

> > ### Author Response · Authors · 2022-08-02
> > **Response to Reviewer C6Jv (2 of 2)**
> >
> >
> > _**Q5 The role of Tranfer Matrix T is not discussed or elaborated.**_
> >
> > **A5**: The role of the transfer matrix $T$ can be explained by Eq. 4 and Figure 1. Formally, with $T$, we can convert the color distribution of the original image $\pmb{x}$ to any specific distribution. For example, in Figure 1, $\pmb{x_H'}$ is mapped via $\pmb{x}$, $\pmb{x_H}$ and $T$.
> >
> >
> > _**Q6 Despite the significant improvement, it is not clear how this proposed method boost the transferability of the adversarial examples.**_
> >
> > **A6**: Compared with existing methods, NCF is more flexible and thus its attack space is larger. Consequently, this helps to search for better adversarial examples. Furthermore, we introduce the Initialization Reset (IR) technique, which helps to jump out of local optimal points. Therefore, the transferability of NCF is better than the existing methods in most cases.
> >
> > _**Q7 About line 129. Why not select a single color distribution as a template from each set? What does the natural representation mean here?**_
> >
> > **A7**: The reviewer seems to misunderstood line 129. For each semantic class, we have 20 clusters (distribution sets) and the style varies from set to set. Since the color distributions in each set are similar, we just select a single color distribution from each set as the template, i.e., using one color distribution to represent each distribution set.
> >
> > As for "natural representation", this means that this is the real distribution sampled from the dataset. Please note that if we average all color distributions of a specific distribution set, resulting color distribution may not be present in the dataset, i.e., fake and unnatural.
> >
> > _**Q8 About line 141. What does the perception is more uniform mean here? How does that help to create adv examples?**_
> >
> > **A8**: Perceptual uniformity means that the pixel space variation is similar to the perception of the human eye, i.e., when the color space value change is large, the human eye perception change should also be large. Conversely, the human eye perceives small changes.
> >
> > There is an underlying assumption in the generation of the adversarial examples: "adversarial examples with small perturbations are less perceptible to the human eye and have higher image quality". In a space where perception is more uniform (e.g., CIELab), it is easier to control the invisibility of the adversarial perturbation [d,e] when the pixel change. However, this assumption does not always hold if in a perceptually less uniform space (e.g., RGB). In this case, even the perturbation of the adversarial example is smaller, it still may be more abrupt to the human eye.
> >
> > [d] Cassidy Laidlaw and Soheil Feizi. Functional adversarial attacks. In NeurIPS, 2019.
> >
> > [e] Zhengyu Zhao, Zhuoran Liu, and Martha A. Larson. Towards large yet imperceptible adversarial image perturbations with perceptual color distance. In CVPR, 2020.

---

> > > ### Comment · Reviewer_C6Jv · 2022-08-10
> > > **Thanks for addressing my concerns**
> > >
> > > Authors have convincingly addressed my concerns and I am willing to increase the score.

---

### Official Review · Reviewer_UgYm · 2022-07-13

**Rating:** 7
**Confidence:** 5
**Soundness:** 2 fair
**Presentation:** 3 good
**Contribution:** 2 fair

**Summary:**

This paper works on Unrestricted color attacks, which manipulate semantically meaningful color of an image.
Current works usually sacrifice the flexibility of the uncontrolled setting to ensure the naturalness of adversarial examples. As a result, the black-box attack performance of these methods is limited. To boost transferability of adversarial examples without damaging image quality, they propose a Natural Color Fool (NCF) which is guided by realistic color distributions sampled from a publicly available dataset and optimized by our neighborhood search and initialization reset. Extensive experiments and visualizations demonstrate the effectiveness of their proposed method.

**Questions:**

1. The proposed method is largely based on a segmentation network and a clustering strategy. What is the impact of different segmentation and clustering algorithms? Also, the clustering is based on the ADE20k dataset. Can this guarantee performance?
2. In the main method, it is unclear to why a random color can successfully attack the classification model with such a high success rate?
3. The main idea seems to be similar to [42]. [42] is a different application, but technically what are their differences?

**Limitations:**

Authors have addressed some limitations and potential negative social impact of their work, what about the feasibility of the attack in the physical world?

**Strengths And Weaknesses:**

The writing and exposition are clear and of high quality. The authors generally address this problem in the style of CV instead of ML.

1. By exploiting the color distribution of semantic classes, the proposed Natural Color Fool (NCF) improves the flexibility of the current unrestricted color attack.
2. Some experimental improvement is obtained.

Problems:
1. The proposed method is mostly empirical without theoretical proof. For example, why the proposed method can improve flexibility?

---

> ### Author Response · Authors · 2022-08-02
> **Response to Reviewer UgYm (1 of 2)**
>
> Thank you for your feedback. We will answer your questions one by one below.
>
> _**Q1 Why the proposed method can improve flexibility?**_
>
> **A1**: Please refer to the response to the generic comment.
>
> _**Q2 What is the impact of different segmentation and clustering algorithms? Also, the clustering is based on the ADE20k dataset. Can this guarantee performance?**_
>
>
> **A2**: Intuitively, the semantic segmentation model, the clustering algorithm and the dataset all have some impact on NCF, but these are not the focus of our paper. So we did not conduct detailed experiments. In this rebuttal, we briefly analyze each part in the following.
>
> Firstly, we compare the performance of our NCF under different semantic segmentation models pre-trained on ADE20K (including Swin-T [a], OCRNet [b] and Deeplabv3+ [c]). As indicated in Table a, segmentation models have impact on the attack success rates of resulting adversarial examples. Among these models, Swin-T is usually the best choice for our NCF. Therefore, in our paper, we choose it to segment inputs. Note that even if the segmentation model affects our method, the lowest black-box attack success rate of NCF is still much higher than the existing methods.
>
> Secondly, intuitively, the better the clustering algorithm, the more obvious the difference in style between clusters, and the larger the search range for semantic classes, then the greater the likelihood of searching for adversarial examples.
>
> Finally, ADE20K contains 150 classes, while other popular semantic segmentation datasets like MS COCO has fewer classes (only 80 semantic classes). Intuitively, the more classes in the dataset, the richer the color distributions library constructed, and the more natural the NCF generated adversarial examples.
>
> Table a: The influence of different segmentation models on attack success rates. (“*” denotes the white-box attack)
>
>
> | Segm       | Res-18*      | VGG-19        | Mobile-v2     | Inc-v3        | Dense-121     | Res-50        | ViT-S         |
> |:----------:|:------------:|:-------------:|:-------------:|:-------------:|:-------------:|:-------------:|:-------------:|
> | Swin-T     | **92.9\*** | **72.1** | **72.7** | **48.3** | **55.3** | **66.7** | 53.0          |
> | OCRNet     | 89.9*        | 69.1          | 67.1          | 44.2          | 50.6          | 61.1          | **56.5** |
> | Deeplabv3+ | 91.0*        | 68.0          | 68.6          | 45.3          | 49.2          | 62.0          | 54.0          |
>
>
> [a] Ze Liu, Yutong Lin, Yue Cao, Han Hu, Yixuan Wei, Zheng Zhang, Stephen Lin, and Baining Guo. Swin transformer: Hierarchical vision transformer using shifted windows. In ICCV, 2021
>
> [b] Yuhui Yuan, Xilin Chen, and Jingdong Wang. Object-contextual representations for semantic segmentation. 2020
>
> [c] Hang Zhang, Chongruo Wu, Zhongyue Zhang, Yi Zhu, Zhi Zhang, Haibin Lin, Yue Sun, Tong He, Jonas Muller, R. Manmatha, Mu Li, and Alexander Smola. Resnest: Split-attention networks. arXiv preprint arXiv:2004.08955, 2020
>
> _**Q3 Why random colors can successfully attack the classification model with such a high success rate?**_
>
> **A3**: NCF-IR-NS (in Table 4) does not mean selecting random colors to attack. Specifically, it first generates a set of adversarial examples with different color distributions and then selects the best example from them based on the loss of the white-box model to attack. Therefore, NCF-IR-NS is close to a white-box attack.
>
> To support our claim, we evaluate the performance of random color attack (NCF-IR-NS-*), i.e., randomly select colors for each semantic class and use the resulting adversarial examples to attack. As demonstrated in Table b, the performance of NCF-IR-NS-\* is much lower than NCF-IR-NS. For example, NCF-IR-NS-\* only achieves a 27.3\% (degraded from 51.6\%) success rate on Inc-v3. Thus, directly using random colors to generate adversarial examples is ineffective.
>
> Table b: The attack success rate of using white-box information and not using it. NCF-IR-NS using Inc-v3 as the substitute model. (“*” denotes the white-box attack)
>
> | Methods     | Inc-v3* | Res-18 | VGG-19 | Mobile-v2 | Dense-121 | Res-50 | ViT-S | XCiT-N12 | DeiT-S |
> |-------------|---------|--------|--------|-----------|-----------|--------|-------|----------|--------|
> | Clean       | 19.2   | 16.1   | 11.4   | 12.8      | 7.9       | 7.5    | 13.3  | 13.7     | 5.8    |
> | NCF-IR-NS   |**51.6***|**43.8**|**42.2**| **42.4**  | **28.0**  |**33.0**|**38.3**| **32.0**|**14.8**|
> | NCF-IR-NS-* | 27.3    | 34.8   | 30.9   | 31.1      | 20.5      | 24.2   | 32.7  | 25.0       | 11.6   |

---

> > ### Author Response · Authors · 2022-08-02
> > **Response to Reviewer UgYm (2 of 2)**
> >
> > _**Q4 The main idea seems to be similar to [42]. [42] is a different application, but technically what are their differences?**_
> >
> > **A4**: No, our main idea is much bigger than [42]. Concretely, 1) the challenge of unrestricted color attack is how to guarantee natural images while large perturbations. We point out that existing solutions lack flexibility and can only perturb in the neighborhood. 2) To overcome this problem, we propose constructing a more flexible color attack space without adjacency in the global space. For this purpose, we introduce [42] to automatically construct the color distribution library. However, the resulting library constructed by [42] is redundant. To simplify it, we select 20 distributions (rather than distribution sets) for each semantic class to represent its color space. 3) Based on our color distribution library, we propose to transfer color distributions for attacks. We further improve the black-box transferability of NCF by proposing IR and NS strategies (see Table 4). Note that without the rest of the NCF, [42] cannot generate effective adversarial examples (see the resulf of NCF-IR-NS-* in Table b).
> >
> > _**Q5 Authors have addressed some limitations and potential negative social impact of their work, what about the feasibility of the attack in the physical world?**_
> >
> >
> > **A5**: We evaluated the attack performance of NCF on Google Cloud Vision API to demonstrate the feasibility of our method in the physical world.
> > First we selected 100 images from the Image-Net-compatible Dataset that were correctly classified on the Google Cloud Vision API. Then NCF was used to generate adversarial examples via the substitute model Res-18. Finally, the resulting adversarial examples are fed into Google Cloud Vision API to perform attack. Notably, NCF can achieve a 42\% attack success rate in this realistic scenario. This shows that NCF is also threatening in real-world applications.

---

### Author Response · Authors · 2022-08-02
**Revision and generic comments (1 of 2)**

We appreciate all reviewers (UgYm, C6Jv and Hmyg) for their insightful comments and are glad to get one "borderline accept" (reviewer UgYm), one "weak accept" (reviewer C6Jv) and one "borderline reject" (reviewer Hmyg) at this time. Encouragingly, all reviewers highlight our extensive experiments on a wide variety of models and the improvement of the attack success rate on the black-box models. UgYm and C6Jv praise for our well-written and easy-to-follow paper. C6Jv highlights our novelty.

We carefully revise the manuscript according to the comments of all the reviewers. For convenience, we highlighted the revised text in color except for the revision of grammars. Here we briefly summarize the updates we have made to the revision:

* cite and discuss the papers the reviewers provided.

* add experiments for the influence of segmentation Models in Appendix H.

* add experiments for the effect of ensemble attack in Appendix I.

* discuss the difference between NCF-IR-NS and random color attack in Appendix J.

---

> ### Author Response · Authors · 2022-08-02
> **Revision and generic comments (2 of 2)**
>
> We have concluded all the comments from all the reviewers and responded to the generic comments as follows:
>
> _**Q1 What does "flexibility" mean? Moreover, why is our approach more flexible than existing methods?**_
>
> **A1**: The "flexibility" in this paper is a relative concept. Existing methods usually have many limitations when modifying the color of an image (as described in Sec 3.2). For example, ReColorAdv requires constraining the perturbation to a relatively small range, which cannot take full advantage of the “unrestricted” setting; cAdv enforces the color belonging to the low-entropy cluster to remain unchanged, which inevitably reduces the attack space; ColorFool manually splits an image into two parts and adds controlled noises on the human-sensitive part, which largely depends on the authors’ intuition (but it varies from person to person). In contrast, NCF does not have these limitations, which automatically makes full use of the “unrestricted” setting.
> Therefore NCF is more flexible than existing methods.

---

### Meta-Review · Area_Chair_dQce · 2022-08-26

**Recommendation:** Accept
**Confidence:** Certain

**Metareview:**

The proposed approach exploits the color distribution of semantic classes, thus improving the flexibility of the current unrestricted color attack. This method generates novel transferrable adversarial attacks. The authors conducted extensive experiments on wide variety of network architectures. A significant improvement of the attack success rate is achieved with the proposed method on both undefended and defended models.

**Award:**

No

---

### Decision · Program_Chairs · 2022-09-14

Accept